# Breastfeeding and Complementary Feeding Practices among Caregivers at Seshego Zone 4 Clinic in Limpopo Province, South Africa

**DOI:** 10.3390/children10060986

**Published:** 2023-05-31

**Authors:** M. H. Mphasha, M. S. Makwela, N. Muleka, B. Maanaso, M. M. Phoku

**Affiliations:** Department of Human Nutrition and Dietetics, University of Limpopo, Polokwane 0727, South Africa

**Keywords:** breastfeeding, exclusive breastfeeding, complementary feeding, children under 24 months, caregivers

## Abstract

Breastfeeding and complementary feeding are key components of infant and young child feeding that ensure healthy growth, survival, and development. Initiating breastfeeding within an hour after delivery, exclusively breastfeeding for six months, and introducing complementary feeding at six months while continuing breastfeeding up to 24 months or beyond, helps in the prevention of malnutrition, which is a public health problem. The aim of this study was to determine breastfeeding and complementary feeding practices among caregivers of children under 24 months in Seshego, Limpopo Province. A quantitative and cross-sectional design was applied to collect data from 86 caregivers using convenience sampling. A structured questionnaire was utilised to gather data and analysed through statistical software, using descriptive and inferential statistics. Chi-square tests were used to determine associations at a 95% confidence interval where a *p*-value of <0.05 was considered statistically significant. The findings show that 55% of participants had good breastfeeding and complementary feeding practices. Moreover, 94.2% of participants breastfeed within an hour after delivery at a healthcare facility. Only 43.6% of children were exclusively breastfed. Most participants (52.3%) gave children food before six months and 45.1% introduced complementary feeding at the appropriate age. Also, 69.7% of children across all age groups were not given infant formula. No statistical association was observed between feeding practices and sociodemographic. Breastfeeding was initiated within an hour after delivery at the healthcare facilities, however, post discharge exclusive breastfeeding maintenance remains a challenge. Few infants were introduced to complementary feeding at the appropriate age. A post discharge intervention to practice exclusive breastfeeding, continued breastfeeding, and introduction of appropriate complementary feeding is recommended.

## 1. Introduction

Breastfeeding is the best way to give infants the nutrition they need for healthy growth and development [1]. Babies’ natural initial food source is breastmilk, which gives them all the energy and nutrients they require for the first six months of life [2]. Breastmilk is easier for a baby’s immature or growing stomach and intestines to digest and contains components that naturally soothe babies. It provides infants with necessary nutrition, supports the growth of their immune systems, and contains the perfect proportion of fat, sugar, water, protein, and vitamins [3]. Breastmilk includes antibodies from the mother that help the body fight diseases [4]. Colostrum, which the yellow or golden milk produced in the first few days, is a vital part of the infant’s diet and immune system [5]. Exclusive breastfeeding (EBF) is very important for a child’s survival and health [2]. Infants who are breastfed exclusively for their first six months of life have decreased infant morbidity and death rates, especially in low- and middle-income nations [6]. Exclusive breastfeeding has numerous benefits in terms of providing energy, protein, water, and other nutrients required for the development of an infant [7].

Breastfeeding and EBF are safe feeding methods that benefit the environment, health, and wellness of mothers, and helps in family planning [8,9]. Breastfeeding and EBF might help speed up recovery after giving birth [10], and in the production of the hormone oxytocin. After delivery, oxytocin will cause the uterus to contract, which helps the vaginal lining return to its normal size and reduces postpartum bleeding [11]. It is also an essential stage in the reproductive process and has significant positive effects on mothers’ health [1] and lowers women’s risk for type 2 diabetes, ovarian cancer, and breast cancer [2]. Moreover, breastfeeding affects sexuality by reducing estrogen levels in women in the early postpartum period [12], which can reduce sexual desire and make having sex uncomfortable for certain women [13]. As a result, this can have an adverse effect on women’s attitudes to breastfeeding. Breastfeeding restricts fathers from helping with the feeding process and bonding with their children, which can cause jealously [14]. Within an hour after the baby’s delivery, breastfeeding or EBF should begin to reduce neonatal mortality and avoid infections. The long-term benefits of breastfeeding for the children are enhanced cognitive development [15] and educational attainment in adulthood [16]. Furthermore, it protects infants against chronic diseases, such as overweight, obesity, and diabetes in their later life [17]. Appropriate breastfeeding could prevent death and save about 823,000 children under the age of five worldwide each year [18]. Promotion, support, and protection of breastfeeding could reduce the risk of death from pneumonia, diarrhea, and neonatal sepsis [3,19]. It can also significantly lower mortality and morbidity from all causes related to general infections, such as gastrointestinal and respiratory tract infections [20,21]. Hence, it is critical to assess feeding practices to evaluate whether children would benefit from these advantages of breastfeeding and EBF.

Exclusive breastfeeding rates vary by nation; however, the worldwide average is 45.7% [22]. A sub-Saharan study showed the prevalence of the early initiation of breastfeeding in West Africa, East Africa, Central Africa, and Southern Africa was 46.94%, 61.82%, 37.84%, and 69.31%, respectively [23]. One of the lowest rates of breastfeeding on the African continent is seen in South Africa [24]. Only 31.6% of infants in South Africa were exclusively breastfed in 2016, with a mean duration of 2.9 months [25]. Most South African children are likely not to benefit from the advantages of breastfeeding and EBF. Studies conducted in South Africa (SA) have shown that most caregivers introduced solid food before 6 months [26,27]. In nations with high infant mortality rates, breastfeeding has been shown to reduce all under-5 deaths by 13%, and using complementary feeding methods appropriately reduces under-5 mortality by an additional 6% [19]. The relatively simple access to infant formula that women and families have had in South Africa has been blamed for the country’s subpar breastfeeding rates, in part because of initiatives to stop the spread of human immunodeficiency virus (HIV) from mothers to children. This policy might have had unfavorable side effects for women who did not have HIV [25]. When breastfeeding in public, women frequently experience embarrassment, stigmatization, and unfavorable attention [28]. Appropriate breastfeeding involves on-demand feeding, which can be time-consuming for women, binding them to the responsibilities of feeding children [28], which compromises mothers’ time to engage on other productive and work responsibilities. Investigating the practices relating to breastfeeding and complementary feeding is critical, and the findings could be used to design an intervention to improve breastfeeding and EBF rates. In addition, the findings will inform future research on the factors impacting appropriate practices.

As infants develop and become more active through the first six months of life, breastmilk is unable to meet all their nutritional requirements; this gap continues to widen as babies and young children age [29,30]. To close these gaps, complementary feeding is essential. Dietary diversity, meal frequency, and timely introduction of extra food constitute good complementary feeding [31,32]. Infants should be actively fed by their caregivers, who should actively encourage eating and pay attention to the child’s hunger signs. Complementary feeding should be timely, adequate, and safe, and the child should be properly fed: timely means that it should be introduced when the baby’s need for energy and nutrients surpasses what can be met through exclusive breastfeeding; adequate means that children get enough energy, protein, and micronutrients to meet the nutritional needs for development; safe foods are those that are prepared and stored hygienically, and are fed with clean hands using clean utensils; and properly fed implies that the food is provided in accordance with a child’s signals of hunger and fullness and with age-appropriate meal frequency and feeding.

The WHO recommends diet diversity in addition to timely introduction to give a broad nutrient intake that meets the developing infant’s demands for all nutrients. This means that a range of the fundamental food categories should be provided as part of the complementary feeding [33]. Between the ages of 6 and 8 months, these complementary foods should be provided at first 2–3 times daily, increasing to 3–4 times daily between the ages of 9 and 11 months, and subsequently 1–2 times daily as desired [33]. Poor complementary feeding practices can stunt a child’s development and exacerbate health issues such as malnutrition, vitamin deficiency, and delayed motor and cognitive development in infant and young children [34]. According to guidelines for feeding infants and young children, iron-rich meals should be introduced to children at the age of six months, and specific foods listed in the South African road to health booklet [35] should be boiled and mashed to make them soft and simple for the baby to swallow [1]. Children should be given a variety of cooked and mashed foods, including starches such as fortified maize meal porridge and potatoes, cooked and mashed vegetables such as butternut and pumpkin, and soft fruits such as bananas and avocados [1,35].

Poor feeding habits, like starting complementary meals too soon or too late and not exclusively breastfeeding for the first six months of life, are a problem in developing nations [35]. In SA, it was found that most caregivers do not practice exclusive breastfeeding and instead give their children food earlier than 6 months [26,27]. However, these studies were not conducted in Seshego where the current study was undertaken. Poor feeding practices are associated with increased risk of malnutrition among children. Malnutrition is a major contributing factor to child morbidity and mortality in developing countries [36]. Malnutrition continues to be a serious public health concern for children under five. It steals children’ dreams and makes their future uncertain [37]. There are several approaches to prevent malnutrition, including starting breastfeeding as soon as feasible, continuing it exclusively, introducing complementary feeding at 6 months, and continuing to breastfeed while providing complementary food until 24 months. Hence, this study seeks to determine breastfeeding and complementary feeding practices among caregivers of children under 24 months in Seshego in the Limpopo Province of South Africa, to understand the breastfeeding and complementary feeding practices there.

## 2. Methodology

### 2.1. Research Design and Setting

A quantitative and cross-sectional survey was adopted for this research to establish the breastfeeding and complementary feeding practices of caregivers of children under 24 months in South Africa. The study was conducted in the Seshego Zone 4 clinic. The clinic offers primary health care services, including postnatal care, to Seshego locals and the public. During postnatal service, breastfeeding and complementary feeding services are also provided to caregivers. 

The township of Seshego is in the Limpopo Province of the Republic of South Africa in the Polokwane Local Municipality of the Capricorn District Municipality. The township is located northwest of Polokwane. There are 8 residential zones in Seshego. According to Statistics South Africa [38], Seshego has a population of 83,863, and 51% are females. In Seshego, 99% of residents are Africans, and 85% of the Africans speak/observe Sepedi culture. There are 24,736 households in Seshego [38].

### 2.2. Population and Sampling of the Study

The target population were caregivers of infants aged 0 to 24 months in Seshego. In this context, the term “caregivers” refers to mothers and/or legal guardians who take on the primary duty of caring for children of the age 0–24 months. Caregivers of infants younger than 24 months were included because their children rely on them for feeding. There were 110 children aged 0–24 months during the study period, which was conducted between October and November 2022. Therefore, a total of 110 constitutes the population size, because the target population was one caregiver for each child under 24 months. Convenience sampling was used to choose caregivers who could speak Sepedi or English and were at least 18 years old so that they could provide consent. A total of 86 caregivers were recruited to participate in this study. The sample size was calculated using the sample calculation formula developed by Krejcie and Morgan [39], which is as follows:$$S=\frac{x^{2}NP\left(1-P\right)}{d^{2}\left(N-1\right)+x^{2}P\left(1-P\right)}$$
where *S* = required sample size; *x*^2^ = the table value of chi-square for 1 degree of freedom at the desired level (3.841); *N* = the population size; *P* = the population proportion (assumed to be 0.50 since this would provide the maximum sample size); and *d* = the degree of accuracy expressed as a proportion (0.05).

### 2.3. Instruments and Data Collection

Data were gathered using a closed-ended questionnaire with two sections: demographic profile, and infant feeding practices. The questionnaire’s infant feeding practices component featured ten questions using a three-point Likert scale (yes; no; not sure), whereas the sociodemographic information section had twelve. The questionnaire was developed using existing literature. Content validity of the instrument was assured using supervisors and dietitians, while reliability of the questionnaire was established through a pilot study. The findings of the pilot study were not included in the overall study and were not published since they informed no changes to the questionnaire. The questionnaire was available in both Sepedi (the primary language spoken by the majority of residents in Seshego) and English (to accommodate the 15% of residents who understand English). The inclusion criteria involved residents who could speak Sepedi and/or understand English. Caregivers who consented were provided with questionnaires to fill out on their own. The questionnaires were filled out by participants in front of the researchers during data collection.

### 2.4. Data Analysis

The Statistical Package for Social Sciences software (SPSS) version 27 was used for the data for analysis. Descriptive statistics were utilized where frequency distributions, means, and standard deviations were computed. Participants’ responses were firstly marked correct or incorrect, which determined the overall practices score. For this study, practice was scored on an overall scale of 100% and classified into 3 categories: poor, fair, and good. Poor practice refers to a total score of 0–50%, fair practice refers to a total score of 51–69%, and good practice refers to a total score of 70–100%. The Chi-squared test was used to determine the relationships between variables, and the level of confidence was set at the 95% confidence interval. A *p*-value of <0.05 was considered statistically significant.

### 2.5. Ethical Issues

The study was approved by the Turfloop Research Ethical Committee (TREC) at the University of Limpopo, which issued clearance certificate number TREC/488/2022: UG. All the study subjects signed a written consent form. Participants were informed that their freedom to leave the study at any moment without consequence was entirely voluntary. Participants’ personal information was also kept confidential.

## 3. Results

### Socio-Demographic Profile

Table 1 shows that most participant caregivers who brought their children to the clinic were of the age group 18–35 years (80.2%), had secondary education or less (55.8%), were unemployed (68.6%), and dependent on social grants (67.4%). Furthermore, the majority were mothers of the children (97.7%), had 2 children or less (60.5%), and 34.0% of the children were age 6 months.

Figure 1 shows that 55% of participants had good feeding practices, followed by 28% with fair practice, and 17% with poor practice.

Table 2 shows that there is no statistically significant difference between all sociodemographic variables and feeding practices.

Table 3 shows that most participants-initiated breastfeeding immediately after birth (94.2%), and 87.2% were still breastfeeding at the time of the study. Most participants were breastfeeding more than 8 times in a day (51.1%), 31.4% of children received medication before 6 months and of those, 59.3% received prescribed medication. Moreover, 43.6% of children were exclusively breastfed, whilst 52.3% gave children food before six months and 45.1% introduced complementary feeding at six months. Also, 69.7% of children were not currently given infant formula.

## 4. Discussions

The purpose of this study was to describe the breastfeeding and complementary feeding practices of children aged of 0 to 24 months among caregivers in Seshego in Polokwane Municipality, Limpopo Province. This study found that 55% of caregivers had good infant feeding practices. These findings are congruent to those of a study conducted by Assefa et al. [40] in Ethiopia, where most mothers engaged in appropriate infant and young child feeding (IYCF) practices. Good IYCF is required to ensure that children grow, develop, and realize their full potential. However, 45% of children were poorly fed, which increases their risk for obesity and chronic diseases later in life. The factors that prevent the accomplishment of good practices must be identified to develop measures for improvement. Most participants (94%) initiated breastfeeding immediately after birth, which is similar to findings of the study by Assefa et al. [40]. Early breastfeeding, which is defined as commencing to breastfeed a baby within an hour after birth, is a high-impact intervention [41], which raises the chances of child survival and provides long-term health benefits [42]. The initiation of breastfeeding within an hour after birth in Burkina Faso is 41%, and lowest prevalence is in Guinea and Kenya at 16.5% and 30.4% respectively [43]. A South African study conducted in Vhembe, Limpopo Province, reported that 67.2% of breastfeeding was initiated within an hour after birth [44]. Shobbo et al. [45], indicated that most of the children who were immediately breastfed after delivery were those born in hospitals, hence the need to increase the number of hospital deliveries. In contrast, evidence from Bangladesh suggests that substandard delivery practices occur in healthcare facilities, which can hinder the early start of breastfeeding [46]. Healthcare facilities should be applauded for promoting appropriate feeding habits for infants and young children. It is concerning, though, that appropriate IYCF practices are not continued after discharge. A post discharge program needs to be created to promote correct IYCF practices. The program should focus on avoiding the introduction of solid food before six months and the use of infant formula.

Only 43.6% of participants in this study exclusively breastfed their children for up to six months. These results are supported by the South African Demographic and Health Survey [47] and a study conducted by Siziba [48], which revealed that 32% and 42% of mothers practice exclusive breastfeeding, respectively. In contrast, a study in Mauritius discovered that only 17.9% of mothers are exclusively breastfeeding [49]. Most participants in the current study were breastfeeding more than 8 times per day (51.1%), in accordance with WHO breastfeeding recommendations that infants under 6 months of age be exclusively breastfed at least 8 times per day [2]. It is rather alarming that 56.4% of infants were not exclusively breastfed for six months of age, which predisposes them to poor development, survival, growth, and disease prevention [50,51]. Even though it is inappropriate to give food or water to infants under the age of six months, 52.3% of children in this study received food. Mushaphi [26] and Shrestha et al. [27] reported similar findings, indicating respectively that 58.3% and 55.6% of caregivers introduced solid food before 6 months. Children who are given food before the recommended six months are at risk of several health problems [32]. According to some research, offering solid meals too soon may raise the risk of dermatitis, type 1 diabetes, obesity, adult-onset celiac disease, and islet autoimmunity, while introducing them too late may exacerbate feeding issues [52,53]. The substitution of the energy- and iron-rich breastmilk with the early introduction to solid foods may potentially raise the risk of diarrheal illness and lead to poor nutritional outcomes, such as insufficient iron reserves [52,54].

It is recommended to refrain from giving infants anything other than breastmilk until they are six months old since it could increase their susceptibility to illness and even threaten life. The disruption of the infant’s feeding pattern may result in decreased breastmilk production, decreased iron absorption from breastmilk, increased risk of infant infections and allergies, and increased risk of obesity [55,56,57,58,59]. It was found that a new pregnancy is also associated with switching too early to complementary feeding [60]. Studies have found that larger percentages of the early introduction of food before the age of six months is a crucial public health concern that requires urgent attention [26,27]. There is a need to strengthen exclusive breastfeeding education using both English and local languages because knowledge and practice are related.

The majority of caregivers in the current study (69.8%) did not give infant formula. This could be attributed to the high cost of infant formulas, since affordability impacts dietary intake [61], including formula feeding. Despite infant formula being inferior to human milk in many ways, this nutritional source promotes more efficient growth, development, and nutrient balance than giving any other food to infants below six months. However, the AFASS (Acceptable, Feasible, Affordable, Sustainable, Safe) principle must be adhered to. The AFASS principles were initially developed to assist mothers who were HIV-positive in making feeding decisions before they start to feed. Mothers should be given the opportunity to select the infant feeding choice that is most suitable for their circumstances after a thorough AFASS assessment has been completed. For the first six months, whether an HIV-positive woman decides to breastfeed or use a replacement feeding method, it is critical that she exclusively breastfeeds to avoid the dangers of mixed feeding [62]. Due to the costs of infant formula and the superiority of breastmilk, exclusive breastfeeding should be advocated for infants below six months. Formula feeding practices are often viewed as the second option to breastmilk and may be influenced by family or cultural practices. One of the biggest barriers to successful breastfeeding is the marketing of breast-milk substitutes, which portrays it as a safe substitute that is equal to or better than human milk. It downplays the benefits and safety of breastfeeding and influences societal norms and the adoption of infant formula use [63]. The International Code on Marketing of Breastmilk Substitutes recommends that nations pass legislation outlawing the marketing of these products [64]. South Africa has already passed this regulation, and the breastmilk substitutes are no longer promoted in some provinces, including Limpopo. Baby foods, which are defined as commercially prepared foods and beverages for infants, are linked to a drop in prolonged breastfeeding and poor IYCF practices [65].

Only 45.1% of caregivers started complementary feeding when children were 6 months old. Baby’s nutritional and energy needs begin to exceed those of breastmilk around six months, necessitating the use of complementary feeding to make up for the gap [66]. However, this study did not establish the types of food eaten and how often are eaten, therefore, it is recommended that future studies on complementary feeding should establish which foods are fed to children.

The limitations of this study include that the results of this investigation showed that infants were being provided food by their caregivers before they were six months old, but they did not specify at what age or when this was happening. Additionally, it was said that fewer children were fed at the recommended age of six months; however, it was not specified which foods were introduced first, how frequently they were fed, or which foods were the most consumed. The data was collected from a small sample; therefore, the findings of this study cannot be generalized to the people residing throughout Polokwane municipality. Some sociodemographic profiles were not included in the chi-square test. The recruitment of participants was done from one clinic, limiting the generalizability of the findings outside of that clinic. Furthermore, participants were being surveyed when their infants were of different ages, which makes the interpretability of some items difficult; therefore, future studies recruiting exclusively caregivers of infants at 6 months of age would make comparability among some of the items more straightforward.

## 5. Conclusions

Breastfeeding is initiated within an hour after delivery at the healthcare facility, however, post discharge continuation of breastfeeding remains a challenge. Most of the children included in this study were not exclusively breastfed and were given food before 6 months of age. Only a few infants were introduced to complementary feeding at the appropriate age. A post discharge intervention to improve breastfeeding, exclusive breastfeeding, and the appropriate introduction of complementary feeding practices is required. It is recommended that a study be conducted that is concerned with the factors that affect the sustainability of exclusive breastfeeding up to six months and the adequate, appropriate, and timely introduction of complementary feeding.

## Figures and Tables

**Figure 1 children-10-00986-f001:**
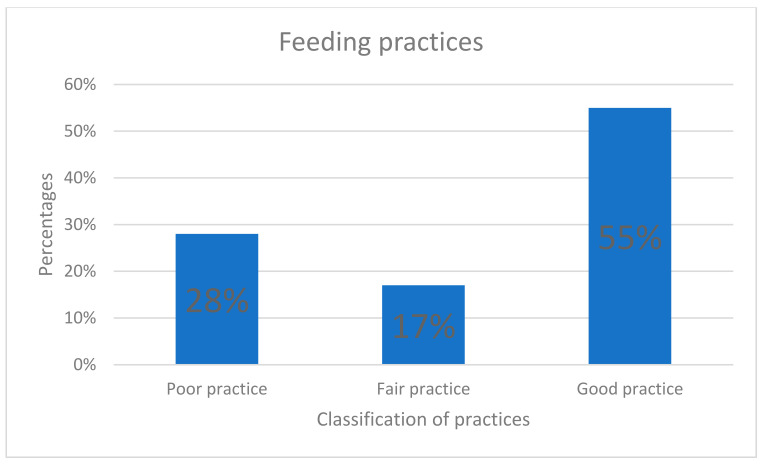
Feeding practices of caregivers.

**Table 1 children-10-00986-t001:** Sociodemographic data of participants.

Variables	Categories	N = 86 (%)
Age of mother/caregiver	18–35 years	69 (80.2%)
	≥36 years	17 (19.8%)
Race	Black	85 (98.8%)
	Coloured	1 (1.2%)
Education status	Secondary education or less	48 (55.8%)
	Tertiary education	38 (44.2%)
Employment status	Temporary employed	7 (8.1%)
	Permanently employed	9 (10.5%)
	Self employed	11 (12.8%)
	Unemployed	59 (68.6%)
Source of income	Social grant	58 (67.4%)
	Pension fund	1 (1.2%)
	Salary	17 (19.8%)
	Other	10 (11.6%)
Number of household members	1–5	55 (63.9%)
	6–10	31 (36.0%)
Relationship to the child	Mother	84 (97.7%)
	Caregiver (grandmother, siblings of mother, nanny)	2 (2.3%)
Number of children	1–2	52 (60.5%)
	3–8	34 (39.5%)
Age of child in months	0–5	15 (17.4%)
	6 months	30 (34.0%)
	7–24 months	41 (47.6%)

**Table 2 children-10-00986-t002:** Overall practices of participants by sociodemographic profile.

Practices of Participants by Sociodemographic Profile		Overall Practices			*p*-Values *
		Poor Practices	Fair Practices	Good Practices	
Age	18–35	12	27	30	0.718
	≥36	1	9	7	
Education	Secondary or less	2	0	2	0.068
	Tertiary education	11	36	35	
Employment	Temporary employed	3	3	1	0.535
	Permanently employed	4	3	2	
	Self employed	6	4	2	
	Unemployed	16	25	17	
Number of children	1–2	8	26	18	0.292
	3–8	5	10	19	
Relationship to the child	Mother	32	30	22	0.349
	Caregiver	0	2	0	

* Signifies statistical significance at the 95% CI.

**Table 3 children-10-00986-t003:** Practices of participants regarding breastfeeding and complementary feeding, N = 86.

Practices of Participants Regarding Breastfeeding and Complementary Feeding	Yes	Not Sure	No
Did you initiate breastfeeding immediately after birth at the healthcare facilities?	81 (94.2%)	1 (1.2%)	4 (4.7%)
Are you still breastfeeding?	75 (87.2%)	0 (0%)	11 (12.8%)
Are you currently giving your child infant formula?	14 (16.4%)	12 (13.9%)	60 (69.7%)
Do you breastfeed 1–7 times in a day?	31 (36.1%)	11 (12.8%)	44 (51.1%)
Do you breastfeed more than 8 times in a day?	44 (51.1%)	11 (12.8%)	31 (36.1%)
Did you exclusively breastfeed your child for up to six months? (*n* = 71)	31 (43.6%)	13 (18.3%)	27 (38.1%)
Have you given the child any medication before 6 months?	27 (31.4%)	0 (0%)	59 (68.6%)
Was the medication prescribed? (*n* = 27)	16 (59.3%)	0 (0%)	11 (40.7%)
Did you give your child food before 6 months?	45 (52.3%)	3 (3.5%)	38 (44.2%)
Did you introduce complementary food at 6 months? (*n* = 71)	32 (45.1%)	11 (15.5%)	28 (39.4%)

*n* = 71 was obtained after subtracting 15 participants with children less than 6 months; *n* = 27 was the number of those who gave their children medication before 6 months.

## Data Availability

This article depends on the data gathered from caregivers of children aged 0–24 months in Seshego township located in Polokwane municipality in Limpopo province of South Africa. The data generated or analysed during the current study is not openly accessible. However, it requested from corresponding author.

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
