# Peer review of "Breastfeeding and Complementary Feeding Practices among Caregivers at Seshego Zone 4 Clinic in Limpopo Province, South Africa"

_children, 2023, doi:10.3390/children10060986_

Round 1

Reviewer 1 Report

Overall comments:

·       The Background contains appropriate information to provide the motivation for the current study. However, it is a bit hard to follow in its current state. In particular, many of the EBF benefits (paragraph 2) and general breastfeeding benefits (paragraph 1) are the same/similar, so paragraph 2 feels duplicative/repetitive after paragraph 1. It may be helpful to integrate those 2 paragraphs and then divide the information into thematic/topical paragraphs, such as EBF/BF benefits to mothers, EBF/BF benefits to mortality, EBF/BF benefits to chronic disease outcomes. Additionally, the information regarding barriers to EBF/BF on page 2, lines 67-76 is very important, but probably belongs in its own paragraph after discussing all of the benefits to EBF/BF.

·       The Background lacks some of the underlying motivation for the current study, particularly what the current study adds to the existing literature. The authors describe the importance of promoting EBF/BF, but what is known about these practices in Sub-Saharan Africa/South Africa specifically is only briefly mentioned, and only with regard to EBF rates and duration. It would be helpful to have a bit more information about what is known about infant feeding habits and barriers to EBF/BF in South Africa, such as the Mushaphi (2008) and Shrestha et al. (2020) references cited in the Discussion. Then, the authors should explain what the current study adds to what is known about EBF/BF beliefs, barriers, and practices in Sub-Saharan Africa/South Africa.

·       The final paragraph of the Background is particularly hard to follow and could benefit from re-organization. It would be helpful to have a bit more information about what is known about infant feeding habits and barriers to EBF/BF in South Africa, such as the Mushaphi (2008) and Shrestha et al. (2020) references from the Discussion (as mentioned above). This would flow well after the description of barriers to EBF/BF (currently on page 2, lines 67-76). Also, the introduction of the current study arrives mid-paragraph (page 3, line 131) after a description of the benefits of good nutrition. I think the section on the motivation of the current study should be its own paragraph, and the line “Caregivers of infants younger than 24 months were included because their children rely on them for feeding.” (page 3, lines 132-133) belongs in the Methodology, not the Background.

·       Because the formula for sampling used the number of children between 0-24 months of age at one particular clinics as the population (page 4, lines 159-162), the authors should exercise caution in the Discussion, as the study was only powered to generalize to the clinic population.

·       Please check the in-text citations against the References carefully. I noticed several inconsistencies that should be corrected.

Specific comments:

·       Page 1, line 10: should be “are key components”

·       Page 1, line 17: should be “convenience sampling”

·       Page 1, lines 18-19: should be “Chi-squared tests were used…”

·       Page 1, lines 21-23: These numbers go back and forth between participants and children. Should these all be participants? For example: “94,2% reported breastfeeding within an hour of delivery…”

·       Page 1, lines 24-26: “Breastfeeding was initiated within an hour 24 after delivery at the healthcare facilities…”: Is this referring to the 94,2% mentioned above? Or another data collection method regarding policy at the institutional level? Please clarify.

·       Page 1, line 26: should be “Few infants were introduced…”

·       Page 1, lines 33-35: I don’t think this citation (Adhikari, et al., 2021) is the correct one for these statements. Do you mean the second citation (WHO) or another, separate citation?

·       Page 2, line 44: should be “…and helps in family planning”

·       Page 2, lines 49-50: Helps the baby’s body fight diseases? Reduces future medical expenditures and prevents potential harm for mothers, babies, or both?

·       Page 2, lines 52-53: This finding needs more context, specifically and increase to what and where. From the original article, this represents projects to “near universal levels” across “75 high-mortality low-income and middle-income countries”, so it’s not quite worldwide.

·       Page 2, line 57: Did you mean “high infant mortality rates”?

·       Page 2, lines 77-93: Many of the findings in this paragraph on the benefits of EBF feel duplicative from the findings described in the first paragraph.  

·       Page 2, lines 77-78: I don’t think this is the correct citation (21) for this information. Please check.

·       Page 2, lines 78-80: I don’t think this is the correct citation (22) for this information. Please check.

·       Page 2, lines 80-82: Several of these citations do not seem to be about under-five mortality. Please check. Also, citation 12 (Lambert et al., 2013) is not here, but could be.

·       Page 2, lines 94-95: The phrase “As infants develop and become more active throughout the first six months of life, breast milk is unable to meet all their nutritional requirements”, and specifically the word “throughout” suggests that complementary feeding is recommended during the first 6 months of life. But later (page 3, lines 129-130 for example), EBF is recommended for the first 6 months of life. Consider rephrasing to clarify, something like, “As infants develop and become more active following the first six months of life…”

·       Page 3, line 118: I don’t think citation 1 is the correct citation for the information provided at the end of this sentence. Please check.

·       Page 3, line 126: I don’t think citation 39 is the correct citation for this information, as this is a qualitative study. Please check.

·       Page 3, lines 127-132: Citations 40 and 41 are the same citation, but one has corrected author affiliations and contributions. Please check the information preceding citations 40 and 41 here to make sure the correct citations are provided for those findings.

·       Page 3, lines 140-141: Please add something like, “of caregivers in South Africa” to clarify the goal of this study.

·       Pages 3-4, lines 143-148: This section switches back and forth between a description of the clinic and a description of Limpopo Province. It would be helpful to describe one first and then the other. It would also be helpful to include a bit more information on Limpopo Province besides the percentage of the population that speaks Sepedi (e.g., race/ethnicity, socio-economic statuses, age, family size, etc.). Additionally, if there is any information that can be provided to describe the population that is served by the clinic, that would also be helpful, such as the demographics of the clinic overall, and/or characteristics of the area (district municipalities or local municipalities, if possible) immediately surrounding the clinic, as Limpopo is quite large, both in population and geography.

·       Page 4, lines 150-158: The line from page 3, lines 132-133 (“Caregivers of infants younger than 24 months were included because their children rely on them for feeding.”) belongs here (around line 152, after the sentence, “In this context, the term "caregivers" refers to mothers and/or legal guardians who take on the primary duty of caring for children of the age 0-24 months.”).

·       Page 4, line 150-158: Please state explicitly here that you recruited 86 caregivers for the study. I know it was the goal, but it would help to clarify that the goal was met.

·       Page 4, line 153: Please add, “there are 110 children aged 0-24 months.”

·       Page 4, lines 159-162: Why was the total number of children aged 0-24 months at this particular chosen as the population size in this formula? Some justification for this decision should be provided, and in the Discussion, caution should be exercised when interpreting the findings, given that the population the study was powered to generalize to was only this particular clinic.

·       Page 4, lines 168-169: Please change to “The questionnaire’s infant feeding practice component featured…”

·       Page 4, lines 172-173: Are the findings of the pilot study published elsewhere? Can you cite them? Was content validity performed first and then the pilot study was conducted? If the pilot study findings are not published elsewhere, can you at least report the sample size of the pilot study, what you assessed, and some broad results, such as the Cronbach's alpha for the infant feeding practices.

·       Page 4, lines 177-178: remove “self-administered by”

·       Page 4, lines 184-188: I’m not clear on how the yes/no/not sure questions were converted to a score out of 100%. Are these the same questions as can be found in Table 3? If so, there are 10 questions there, not 9 as described in lines 168-169. Can you describe this scoring here?

·       Page 5, lines 198-199: Please change to “were unemployed”

·       Page 5, line 203: Please add “(n = 86)” to the table title.

·       Page 5, line 203: I checked all of these percentages, and I believe only 2 require correction: the percentage for 1-5 household members should be “64,0%” and the percentage for 6 months under age of child in months should be “34,9%”.

·       Page 6, line 213: Please add “(n = 86)” to the table title.

·       Page 6, line 214: Can you make your categories in this table match those in Table 1? Or at least combine categories from Table 1 into this table? In Table 2, under Education, your categories are “Primary or less” and then “Secondary or more” but in Table 1 your categories are “Secondary education or less” and “Tertiary education”. I can’t reconcile those 2 tables’ values for those different education categories, so please check the values and make the categories consistent or at least subcategories or the other table’s categories.

·       Page 6, line 214: Additionally, the numbers for the Education category here do not sum to 86, they sum only to 85. The numbers from Table 1 suggest these values are not missing, as those values summed to 86, so please check these numbers. If some participants did not respond to this question, please add a parenthetical “(n = 85)” next to “Education” along with a table footnote explaining that values do not sum to 86 due to missing data.

·       Page 6, line 214: Also in Table 2, the mother/caregiver numbers here do not match the values in Table 1. In Table 1 the authors reported 84 mothers and 2 caregivers, but in Table 2 the authors reported 14 mothers and 72 caregivers. Please correct this.

·       Page 6, lines 217-219: Change to “31,5% of children received medication before 6 months and of those, 59,3% received prescribed medication.

·       Page 6, line 218: This number is 31,5% but the number in Table 3 is 31,4%. I believe 31,5% is the correct number, but these values should be checked.

·       Page 6, lines 221-222: The question as written in Table 3 seems to be asking the participants if they are currently feeding their children formula. So I think this sentence should be rephrased to reflect that: “Also 69,7% of children were not currently being given infant formula.”

·       Pages 6-7, lines 224-226 (Table 3): There are several rounding errors in this table, including the one mentioned above where there is a discrepancy between the value mentioned in text vs. the same value in Table 3 (31,5% vs. 31,4%), so please check all of these percentages.

·       Pages 6-7, lines 224-226 (Table 3): There are a few percentages written with periods for decimals rather than commas. Please make this consistent.

·       Pages 6-7, lines 224-226 (Table 3): The sum of all frequencies for the final item “Did you introduce complementary food at 6 months? (n=27)” add up to 71, not 27. Based on the Table 3 note and the percentage values, I think it’s the parenthetical (n=27) that needs to be changed to (n=71).

·       Pages 6-7, lines 224-226 (Table 3): The Table 3 note has an asterisk, but there are no asterisks that appear in the table itself. These should be added.

·       Page 7, lines 231-238: I know it was mentioned in its first citation (lines 231-232) that the Assefa et al. study was conducted in Ethiopia, but it may be helpful to name the study as the Assefa et al. study when it is first cited so that when it is mentioned again (lines 237-238), it is more obvious that this finding comes from a study conducted in Ethiopia.

·       Page 7, lines 241-242: Where does this finding come from? And can you name the countries where those values come, and possibly the South African value if it is available so readers can contextualize the current study’s findings?

·       Page 7, lines 242-245: This is a run-on sentence that should be edited.

·       Page 7, lines 245-247: It should be mentioned that this evidence comes from a different global region (i.e., Bangladesh).

·       Page 7, lines 247-250: If I am interpreting the results from Table 3 correctly, the main barrier to EBF for 6 months is giving children food in addition to breastfeeding before 6 months of age, given that most participants were still breastfeeding (n = 75, 87,2%) and few respondents reported giving their children formula (n = 14, 16,3%). In discussing promoting appropriate feeding habits for infants and young children at this clinic, then, a post-discharge program should focus on these particular barriers to EBF for 6 months.

·       Page 7, lines 252-253: This number should be 43,7%, both here and in Table 3 itself.

·       Page 7, line 254: “Siziba” should be something like, “a study conducted by Siziba et al.”

·       Page 7, lines 253-255: The first line of the Results section of the abstract in the Sizba et al. study is “The EBF rate for infants up to the age of six months was 12%.” So should this 42% number actually be 12%, and the findings in the current study demonstrate higher rates of EBF through 6 months of age than have been previously reported?

·       Page 7, lines 255-256: Add that this is EBF for the first 6 months.

·       Page 7, lines 256-259: Shouldn’t this 59% actually be 51,1%, according to Table 3?

·       Page 7, lines 256-259: Citation number 1 is not a WHO reference. Please check this citation.

·       Page 7, lines 259-260: Include that this is “not exclusively breastfed for the first six months of age”.

·       Page 7, lines 259-260: Where is “more than 57%” coming from? Is this the Not Sure responses plus the No responses? That is 56,3%.

·       Page 8, lines 262-263: According to the reference section, Mushaphi is not an et al. reference, but Shrestha should be Shrestha et al.

·       Page 8, lines 264-265: Please check to ensure this is the correct citation for this finding. That study seems to be a description of infant feeding practices in Abu Dhabi, UAE, and not a study on health problems observed in children fed food before the recommend age of six months.

·       Page 8, lines 274-277: I am having a hard time following the end of this sentence. Does “a new pregnancy” mean risk of interpregnancy intervals of less than 18 months? Can that be stated more explicitly? A new pregnancy in and of itself would not otherwise fit into this list of risks.

·       Page 8, lines 277-279: “These findings” means findings from the current study? Or the findings described in the prior sentence?

·       Page 8, lines 280-281: This argument regarding the need for EBF education should be better contextualized, as the current study did not examine EBF education or participant awareness of EBF recommendations/education. If this recommendation comes from other studies and the current study underscores that optimal infant feeding is still an issue, then make sure that’s clear.

·       Page 8, line 282: This should be 69,8% or nearly 70%.

·       Page 8, lines 282-284: Is this this right citation for this finding? That study interviewed nurses, not mothers.

·       Page 8, lines 294-295: I am not sure I am following the logic of this sentence. Is the cost of infant formula really one of the 2 main reasons for why EBF should be advocated for?

·       Page 8, lines 300-302: You state this recommendation, but not whether or not South Africa has passed such legislation or not, and if not, whether such practices persist in Limpopo Province.

·       Page 8, lines 305-306: Isn’t this figure 45,1% (from Table 3)?

Page 8, lines 322-328: Additional limitations should be included, such as the recruitment of all participants from one clinic, limiting the generalizability of the findings outside of that clinic; participants being surveyed when their infants were different ages, making the interpretability of some items difficult, and future studies recruiting exclusively caregivers of infants at 6 months of age would make comparability among some of the items more straightforward.

Author Response

Reviewer comment

Implementation

Page

However, it is a bit hard to follow in its current state. In particular, many of the EBF benefits (paragraph 2) and general breastfeeding benefits (paragraph 1) are the same/similar, so paragraph 2 feels duplicative/repetitive after paragraph 1. It may be helpful to integrate those 2 paragraphs and then divide the information into thematic/topical paragraphs, such as EBF/BF benefits to mothers, EBF/BF benefits to mortality, EBF/BF benefits to chronic disease outcomes. Additionally, the information regarding barriers to EBF/BF on page 2, lines 67-76 is very important, but probably belongs in its own paragraph after discussing all of the benefits to EBF/BF.

The paragraphs have been merged, re-segmented and reorganized.

Page 1-3, line 34-137

The authors describe the importance of promoting EBF/BF, but what is known about these practices in Sub-Saharan Africa/South Africa specifically is only briefly mentioned, and only with regard to EBF rates and duration. It would be helpful to have a bit more information about what is known about infant feeding habits and barriers to EBF/BF in South Africa, such as the Mushaphi (2008) and Shrestha et al. (2020) references cited in the Discussion. Then, the authors should explain what the current study adds to what is known about EBF/BF beliefs, barriers, and practices in Sub-Saharan Africa/South Africa.

Done as recommended

Page 2, line 77-78

Caregivers of infants younger than 24 months were included because their children rely on them for feeding.” (page 3, lines 132-133) belongs in the Methodology, not the Background.

Taken to methodology

Page 4, line 155-156

Because the formula for sampling used the number of children between 0-24 months of age at one particular clinics as the population (page 4, lines 159-162), the authors should exercise caution in the Discussion, as the study was only powered to generalize to the clinic population.

Caution exercised

Page 7-9, line 233-332

Please check the in-text citations against the References carefully. I noticed several inconsistencies that should be corrected.

Checked and aligned for consistency

Page 1-9, line 33-316

Page 1, line 10: should be “are key components”

Done

Page 1, line 10

Page 1, line 17: should be “convenience sampling

Done

Page 1, line 17

Page 1, lines 18-19: should be “Chi-squared tests were used…”

Done

Page 1, line 18-19

Page 1, lines 24-26: “Breastfeeding was initiated within an hour 24 after delivery at the healthcare facilities…”: Is this referring to the 94,2% mentioned above? Or another data collection method regarding policy at the institutional level? Please clarify.

Clarified

Page 1, line 24-26

Page 1, line 26: should be “Few infants were introduced…”

Done

Page 1, line 26

Page 1, lines 33-35: I don’t think this citation (Adhikari, et al., 2021) is the correct one for these statements. Do you mean the second citation (WHO) or another, separate citation?

Corrected

Page 1, line 33-35

Page 2, line 44: should be “…and helps in family planning

Done

Page 2, line 44

Page 2, lines 49-50: Helps the baby’s body fight diseases? Reduces future medical expenditures and prevents potential harm for mothers, babies, or both?

Rephrased

Page 2, line 49-50

Page 2, lines 52-53: This finding needs more context, specifically and increase to what and where. From the original article, this represents projects to “near universal levels” across “75 high-mortality low-income and middle-income countries”, so it’s not quite worldwide.

Contextualized

Page 2, line 52-53

Page 2, line 57: Did you mean “high infant mortality rates”?

Yes and clarified

Page 2, line 79

Page 2, lines 77-93: Many of the findings in this paragraph on the benefits of EBF feel duplicative from the findings described in the first paragraph. 

Merged in paragraph 1 of background

Page 1-2, line 33-70

Page 2, lines 77-78: I don’t think this is the correct citation (21) for this information. Please check.

Citation corrected

Page 1, line 42-43

Page 2, lines 78-80: I don’t think this is the correct citation (22) for this information. Please check.

Citation corrected

Page 2, line 44-45

Page 2, lines 94-95: The phrase “As infants develop and become more active throughout the first six months of life, breast milk is unable to meet all their nutritional requirements”, and specifically the word “throughout” suggests that complementary feeding is recommended during the first 6 months of life. But later (page 3, lines 129-130 for example), EBF is recommended for the first 6 months of life. Consider rephrasing to clarify, something like, “As infants develop and become more active following the first six months of life…”

Rephrased

Page 3, line 95-97

Page 3, line 118: I don’t think citation 1 is the correct citation for the information provided at the end of this sentence. Please check.

Corrected

Page 3, line 117-119

Page 3, line 126: I don’t think citation 39 is the correct citation for this information, as this is a qualitative study. Please check.

Corrected

Page 3, line 128-129

Page 3, lines 127-132: Citations 40 and 41 are the same citation, but one has corrected author affiliations and contributions. Please check the information preceding citations 40 and 41 here to make sure the correct citations are provided for those findings.

Corrected

Page 3, line 133-134

Page 3, lines 140-141: Please add something like, “of caregivers in South Africa” to clarify the goal of this study.

Done

Page 4, 140-141

Pages 3-4, lines 143-148: This section switches back and forth between a description of the clinic and a description of Limpopo Province. It would be helpful to describe one first and then the other. It would also be helpful to include a bit more information on Limpopo Province besides the percentage of the population that speaks Sepedi (e.g., race/ethnicity, socio-economic statuses, age, family size, etc.). Additionally, if there is any information that can be provided to describe the population that is served by the clinic, that would also be helpful, such as the demographics of the clinic overall, and/or characteristics of the area (district municipalities or local municipalities, if possible) immediately surrounding the clinic, as Limpopo is quite large, both in population and geography.

Revised

Page 4, line 140-151

·       Page 4, line 150-158: Please state explicitly here that you recruited 86 caregivers for the study. I know it was the goal, but it would help to clarify that the goal was met.

Stated

Page 4, line 161

Page 4, line 153: Please add, “there are 110 children aged 0-24 months

Page 4, lines 159-162: Why was the total number of children aged 0-24 months at this particular chosen as the population size in this formula? Some justification for this decision should be provided, and in the Discussion, caution should be exercised when interpreting the findings, given that the population the study was powered to generalize to was only this particular clinic.

Justified

Page 4, line 159

Page 4, lines 168-169: Please change to “The questionnaire’s infant feeding practice component featured…”

Done

Page 4, line 171-172

Page 4, lines 177-178: remove “self-administered by”

Removed

Page 4, lines 184-188: I’m not clear on how the yes/no/not sure questions were converted to a score out of 100%. Are these the same questions as can be found in Table 3? If so, there are 10 questions there, not 9 as described in lines 168-169. Can you describe this scoring here?

Described

Page 3, line 186-187

Page 5, lines 198-199: Please change to “were unemployed

Done

Page 5, line 205

Page 5, line 203: Please add “(n = 86)” to the table title

Added

Page 5, line 209

Page 6, line 214: Can you make your categories in this table match those in Table 1? Or at least combine categories from Table 1 into this table? In Table 2, under Education, your categories are “Primary or less” and then “Secondary or more” but in Table 1 your categories are “Secondary education or less” and “Tertiary education”. I can’t reconcile those 2 tables’ values for those different education categories, so please check the values and make the categories consistent or at least subcategories or the other table’s categories.

Aligned

Page 5 & 6, line 209 & 220

Page 6, lines 217-219: Change to “31,5% of children received medication before 6 months and of those, 59,3% received prescribed medication.

Done

Page 6, line 224

Page 6, line 218: This number is 31,5% but the number in Table 3 is 31,4%. I believe 31,5% is the correct number, but these values should be checked.

Corrected

Page 7, line 229

Page 6, lines 221-222: The question as written in Table 3 seems to be asking the participants if they are currently feeding their children formula. So I think this sentence should be rephrased to reflect that: “Also 69,7% of children were not currently being given infant formula.”

Done

Page 7, line 229

Pages 6-7, lines 224-226 (Table 3): There are several rounding errors in this table, including the one mentioned above where there is a discrepancy between the value mentioned in text vs. the same value in Table 3 (31,5% vs. 31,4%), so please check all of these percentages.

Checked and corrected

Page 7, line 229

Pages 6-7, lines 224-226 (Table 3): There are a few percentages written with periods for decimals rather than commas. Please make this consistent.

Commas used for consistency

Page 7, line 229

Pages 6-7, lines 224-226 (Table 3): The sum of all frequencies for the final item “Did you introduce complementary food at 6 months? (n=27)” add up to 71, not 27. Based on the Table 3 note and the percentage values, I think it’s the parenthetical (n=27) that needs to be changed to (n=71).

Changed

Page 7,line 229

Pages 6-7, lines 224-226 (Table 3): The Table 3 note has an asterisk, but there are no asterisks that appear in the table itself. These should be added.

Added

Page 7, line 229

Page 7, lines 231-238: I know it was mentioned in its first citation (lines 231-232) that the Assefa et al. study was conducted in Ethiopia, but it may be helpful to name the study as the Assefa et al. study when it is first cited so that when it is mentioned again (lines 237-238), it is more obvious that this finding comes from a study conducted in Ethiopia.

Done

Page 7, line 237-238

Page 7, lines 241-242: Where does this finding come from? And can you name the countries where those values come, and possibly the South African value if it is available so readers can contextualize the current study’s findings?

Country named

Page 7, lines 242-245: This is a run-on sentence that should be edited.

Edited

Page 7, line 250-253

Page 7, lines 245-247: It should be mentioned that this evidence comes from a different global region (i.e., Bangladesh).

Done

Page 7, line 251

age 7, lines 247-250: If I am interpreting the results from Table 3 correctly, the main barrier to EBF for 6 months is giving children food in addition to breastfeeding before 6 months of age, given that most participants were still breastfeeding (n = 75, 87,2%) and few respondents reported giving their children formula (n = 14, 16,3%). In discussing promoting appropriate feeding habits for infants and young children at this clinic, then, a post-discharge program should focus on these particular barriers to EBF for 6 months.

Done

Page 7, line 253-258

Page 7, line 254: “Siziba” should be something like, “a study conducted by Siziba et al.”

Done

Page 8,line 262

Page 7, lines 255-256: Add that this is EBF for the first 6 months.

Done

Page 8, line 267

Page 7, lines 256-259: Citation number 1 is not a WHO reference. Please check this citation.

Checked and corrected

Page 8, line 266

Citations in discussions should be corrected since some not correct

Done

Page 7-9, line 234-332

Page 8, lines 300-302: You state this recommendation, but not whether or not South Africa has passed such legislation or not, and if not, whether such practices persist in Limpopo Province.

Stated

Page 8, line 310-311

Page 8, lines 322-328: Additional limitations should be included, such as the recruitment of all participants from one clinic, limiting the generalizability of the findings outside of that clinic; participants being surveyed when their infants were different ages, making the interpretability of some items difficult, and future studies recruiting exclusively caregivers of infants at 6 months of age would make comparability among some of the items more straightforward.

Added and paragraph of limitations taken to last paragraph of discussion as per reviewer 2’s recommendation

Page 9, line 323-332

Reviewer 2 Report

Comments and suggestions are attached below. This is an interesting study to explain the important of breastfeeding and complementary feeding practices among caregivers. However, some correction need to be done to improve the interpretation specifically in drawing discussion and conclusion.

Author Response

Reviewer comment

Implementation

Page

Revise abstract to include statistical analysis results

Done

Page 1, line 17-19

Divide first paragraph of abstract into two.

Done

Page 1-2, line 33-70

Merging research design and setting

Done

Page 3-4, line 139-151

Add limitations and strength of the study in the last paragraph.

Done

Page 9, line 232-233
